# Genomic Inbreeding and Runs of Homozygosity Analysis of Cashmere Goat

**DOI:** 10.3390/ani14081246

**Published:** 2024-04-22

**Authors:** Qian Zhao, Chang Huang, Qian Chen, Yingxiao Su, Yanjun Zhang, Ruijun Wang, Rui Su, Huijuan Xu, Shucai Liu, Yuehui Ma, Qianjun Zhao, Shaohui Ye

**Affiliations:** 1Department of Animal Breeding and Reproduction, College of Animal Science and Technology, Yunnan Agricultural University, Kunming 650201, China; zhaoqian12242021@163.com (Q.Z.); brookhuang2000@163.com (C.H.); 2State Key Laboratory of Animal Biotech Breeding, Institute of Animal Sciences, Chinese Academy of Agricultural Sciences (CAAS), Beijing 100193, China; chenqian602141@163.com (Q.C.); suyingxiao00@163.com (Y.S.); mayuehui@caas.cn (Y.M.); 3College of Animal Science, Inner Mongolia Agricultural University, Hohhot 010018, China; imauzyj@163.com (Y.Z.); nmgwrj@126.com (R.W.); suruiyu@126.com (R.S.); 4Chifeng Hanshan White Cashmere Goat Breeding Farm, Chifeng 024506, China; xhj8669@126.com (H.X.); lsc9892@126.com (S.L.)

**Keywords:** cashmere goat, runs of homozygosity, inbreeding coefficient, genetic diversity, selection signatures

## Abstract

**Simple Summary:**

Cashmere goats are valuable genetic resources which are famous for their high-quality fiber. Runs of homozygosity (ROHs) refer to uninterrupted sequences of homozygous genotypes present within the DNA sequence of an individual. To enhance the preservation, improvement and enduring utilization of this valuable genetic resource, we performed a comprehensive study on the genetic variance, degree of inbreeding, patterns of ROHs and selected genes situated in ROH islands which are associated with economic attributes in goats via whole-genome sequencing analysis. Our research showed that the Inner Mongolia cashmere goat possesses a notably high level of genetic diversity and presented the lowest inbreeding coefficient. On the contrary, the highest $F_{roh}$ and lowest levels of diversity were observed in wild goats compared to domesticated goats. The analysis of selection signal through ROHs identified some genes related to meat, fiber and milk production; fertility, disease and cold resistance and adaptability; and body size and growth.

**Abstract:**

Cashmere goats are valuable genetic resources which are famous worldwide for their high-quality fiber. Runs of homozygosity (ROHs) have been identified as an efficient tool to assess inbreeding level and identify related genes under selection. However, there is limited research on ROHs in cashmere goats. Therefore, we investigated the ROH pattern, assessed genomic inbreeding levels and examined the candidate genes associated with the cashmere trait using whole-genome resequencing data from 123 goats. Herein, the Inner Mongolia cashmere goat presented the lowest inbreeding coefficient of 0.0263. In total, we identified 57,224 ROHs. Seventy-four ROH islands containing 50 genes were detected. Certain identified genes were related to meat, fiber and milk production (*FGF1*, *PTPRM*, *RERE*, *GRID2*, *RARA*); fertility (*BIRC6*, *ECE2*, *CDH23*, *PAK1*); disease or cold resistance and adaptability (*PDCD1LG2*, *SVIL*, *PRDM16*, *RFX4*, *SH3BP2*); and body size and growth (*TMEM63C*, *SYN3*, *SDC1*, *STRBP*, *SMG6*). 135 consensus ROHs were identified, and we found candidate genes (*FGF5*, *DVL3*, *NRAS*, *KIT*) were associated with fiber length or color. These findings enhance our comprehension of inbreeding levels in cashmere goats and the genetic foundations of traits influenced by selective breeding. This research contributes significantly to the future breeding, reservation and use of cashmere goats and other goat breeds.

## 1. Introduction

Runs of homozygosity (ROHs) refer to uninterrupted sequences of homozygous genotypes present within the DNA sequence of an individual. These segments are identical because they have been inherited from both parents and their ancestors sharing a common genetic background [1]. In general, a shorter ROH indicates common ancestors in the descent, while a longer ROH suggests that recent inbreeding has occurred, reflecting more immediate common ancestry and genetic homogeneity due to mating between close relatives [2]. ROHs are useful for understanding the genetic connections among individuals and aiding in the reduction in inbreeding rates and the discovery of harmful genetic variations [1]. Analyzing the ROH pattern in the genome, it is possible to explore the inbreeding level, genetic diversity and genetic defects in livestock and poultry. Utilizing ROHs to calculate the inbreeding coefficient ($F_{ROH})$ can precisely assess autozygosity and is more effective in identifying both historical and recent inbreeding than pedigree-based estimates ($F_{Ped}$) [3]. Increasing evidence indicates $F_{ROH}$ can be used as an indicator for a population’s inbreeding levels, especially in situations where pedigree data are not available [4,5,6,7].

In recent years, due to the extensive utilization of SNP microarray gene chip and whole-genome resequencing technologies [8,9,10,11], research into ROHs based on livestock and poultry genomic information has been increasing. This includes research on various species, such as goats [12,13,14], sheep [15,16,17], cattle [7,18,19], pigs [3,20,21], chickens [22,23,24] and so on. Genomic regions that were abundant with ROHs were defined as ROH islands, potentially signaling selection sweeps. For instance, in cattle, pigs and sheep, some ROH islands have been discovered to correlate with genes related to reproduction, immune function and local adaptability [13,16,18,20]. ROHs also can be utilized as a useful tool to evaluate genetic diversity and inbreeding, providing insights into both recent and ancient inbreeding through the analysis of short and long ROH segments, respectively [25].

Inner Mongolia cashmere goats and Hanshan White cashmere goats are outstanding Chinese cashmere goat breeds that adapt well to temperate continental semi-arid and cold climates. These breeds are known for their thin fiber diameter and abundant cashmere production, which play important roles in the global high-quality production of cashmere for the wool market. Current research primarily focuses on differential gene expression, key genes related to cashmere traits and molecular regulatory mechanisms of hair follicle development during the cashmere growth cycle [26,27,28]. Few studies suggest that inbreeding has a significant impact on the cashmere traits of Inner Mongolia cashmere goats [29,30]. However, information about ROH patterns and inbreeding levels in Inner Mongolia cashmere goats and Hanshan White cashmere goats is scarce.

In our study, we aimed to (1) characterize genome-wide ROH patterns and inbreeding levels; and (2) identify ROH islands and putative candidate genes associated with the cashmere trait in cashmere goats based on resequencing data from Inner Mongolia cashmere goats, Hanshan White cashmere goats and other non-cashmere goats. This study is of significant importance for the future breeding, conservation and utilization of goat breeds such as the Inner Mongolia cashmere goat and Hanshan White cashmere goat.

## 2. Materials and Methods

### 2.1. Experimental Population, DNA Extraction and Whole-Genome Resequencing

In this research, a total of 123 goats (including 113 domestic goat individuals and 10 wild goat individuals) were analyzed, wherein the ear tissue of 55 Inner Mongolia Erlangshan cashmere goat (IMC) and Hanshan White cashmere goat (HSC) individuals was collected and conserved in 75% ethyl alcohol for DNA extraction. Additionally, we downloaded 68 individual goats’ genome resequencing data from the database of NCBI https://www.ncbi.nlm.nih.gov (accessed on 23 March 2024). The sample populations included Inner Mongolia cashmere goats (IMC, *n* = 51), Hanshan White cashmere goats (HSC, *n* = 15), Alpine goats (ALG, *n* = 10), Boer goats (BOE, *n* = 12), Jining Gray goats (JNG, *n* = 11), Saanen Dairy goats (SAA, *n* = 14) and a wild goat population (Siberian Ibex) (IBE, *n* = 10). Details on sample names, data sources, abbreviations, geographical distributions, and numbers are provided in Table 1 and Appendix A.

The Inner Mongolia cashmere goat is divided into three types based on central production: the Albas type, the Erlangshan type and the Alxa type. Since the 1960s, selective breeding and cooperative breeding have been carried out for this breed. The Hanshan White cashmere goat is a breed developed from local goats and Liaoning cashmere goats. The center of the production of Hanshan White cashmere goats is located in Balin Right Banner, Chifeng City, Inner Mongolia.

Genomic DNA was extracted by employing a Wizard^®^ Genomic DNA Purification Kit (Promega, Madison, WI, USA), adhering to the guidelines provided by the manufacturer. DNA concentration was determined by using a Nanodrop1000 spectrophotometer (Thermo Fischer Scientific, Wilmington, DE, USA). DNA exhibiting a purity ratio (A260/280) of 1.8 to 2.0 and ≥20 ng/μL was utilized. Following the guidelines provided by the manufacturer (Illumina Lnc., San Diego, CA, USA), sequencing libraries were constructed and sequenced using the DNBSEQ-T7 platform (Shenzhen MGI Co., Ltd., Shenzhen, China) with PE150.

### 2.2. The Processing of Whole-Genome Sequencing Data

Raw data were trimmed for quality using fastp (version 0.20.1) (https://github.com/OpenGene/fastp, accessed on 23 March 2024), and the quality of the trimmed reads was assessed with FastQC (http://www.bioinformatics.babraham.ac.uk/projects/fastqc, accessed on 23 March 2024). After filtering, the remaining reads were aligned to the reference goat genome (ARS1 1.2, GCA_001704415.1) using the Burrows–Wheeler Aligner (BWA, version 0.7.17) [31]. Next, bam files were sorted and indexed utilizing samtools (Samtools-1.7) [32]. Picard software (https://broadinstitute.github.io/picard/, accessed on 23 March 2024) was utilized to remove PCR repeats. After, Genome Analysis Toolkit software (GATK, version 4.1.8.0) (https://software.broadinstitute.org/gatk, accessed on 23 March 2024) was utilized. Haplotype Caller was employed to generate GVCF files, and multiple gVCF files were combined into one VCF file using GATK GenomicsDBImport and GenotypeGVCFs. Then, we used GATK Variantfiltration to remove low-quality variant sites.

Next, PLINK v1.9 (https://www.cog-genomics.org/plink, accessed on 23 March 2024) [33] was used, excluding SNPs with a missingness rate greater than 0.1 (–geno 0.1) with a minor allele frequency (MAF) less than 5% (–maf 0.05) that departed from the Hardy–Weinberg equilibrium at *p*  <  10^−3^ (–hwe 0.001). Finally, only autosomal SNPs were retained for additional analysis. Individuals presenting over 10% missing genotypes (–mind 0.1) were excluded. For the analysis of ROH, the files were quality-controlled without setting the minimum allele frequency (–maf) parameter [34]. The remaining SNPs were annotated according to their positions using SnpEff v4.35 [35].

### 2.3. The Analysis of Population Structure and Linkage Disequilibrium

Principal component analysis (PCA) was conducted using GCTA software (version 1.25.3) [36] (–make-grm, –grm) to assess the genetic structure. We used PLINK v1.9 to calculate the genetic distance matrix (–distance matrix) and applied the a‘pe’ package in R to construct a neighbor-joining (NJ) tree, which was visualized on the iTOL website. Model-based clustering to refine the population structure was fulfilled using the Admixture (v1.3.0, http://dalexander.github.io/admixture/download.html, accessed on 23 March 2024). For K = 1 to 7, the number of clusters (K values) for all samples was determined. The best numbers of K clusters were determined by the minimum cross-validation (CV) error rate. In this study, linkage disequilibrium (LD) was measured with PLINK, and the correlation coefficients (r²) of alleles were calculated by PopLDdecay software (https://github.com/BGI-shenzhen/PopLDdecay, accessed on 23 March 2024) [37].

### 2.4. Identification of ROH

We used PLINK software (version 1.90) with the “–homozyg” command to identify ROHs. The main parameter settings were as follows: each ROH segment needed to contain at least 30 SNPs and have a minimum length of 200 kb, a density standard of one SNP per 30 kb and a maximal interval of 500 kb between two consecutive SNPs within an ROH; the sliding window size was 50 SNPs, permitting up to one heterozygous site and five missing sites within each window; and the sliding window threshold was set to 0.05 (–homozyg-window-snp 50, –homozyg-snp 30, –homozyg-kb 200, –homozyg-density 30, –homozyg-gap 500, –homozyg-window-missing 5, –homozyg-window-threshold 0.05, –homozyg-window-het 1).

### 2.5. ROH Classification and Assessment of Inbreeding Coefficients

Based on ROH segment length, we classified ROHs into three groups: small (<300 kb), medium (300 kb–1.5 Mb) and large (>1.5 Mb). The count of ROH in each length category and the ROH number of each chromosome was determined across seven goat populations. The total number and length of ROHs for each animal was also assessed. $F_{ROH}$ for each individual were determined using the equation proposed by McQuillan et al. [38]: =Σ $L_{roh}/L_{auto}$. Here, $L_{roh}$ represents the cumulative length of all ROHs in an individual genome, and $L_{auto}$ is the total length of the autosomes covered by SNPs, which was 2466.19 Mb in our study.

### 2.6. Identification of Genes within ROH Islands of Domestic Goat Populations

In our study, the incidence of ROH was calculated as the ratio of animals within domestic goat populations that possess an SNP located in an ROH segment. This analysis was visualized through Manhattan plots, generated utilizing the qqman package (version 0.1.9). We established the top 0.1% of SNPs within ROHs as the threshold for identifying the genomic areas most frequently related to ROHs across each population. A contiguous string of neighboring SNPs exceeding this threshold constituted what we have termed an ROH island, delineating genomic regions significantly impacted by ROHs. The specific selection method for the top 0.1% of the SNPs is based on the research content of Gorssen [39]. The genes of the ROH islands were annotated using SnpEff v4.35. David (https://david.ncifcrf.gov, accessed on 23 March 2024) was utilized to achieve Gene Ontology (GO) and Kyoto Encyclopedia of Genes and Genomes (KEGG) pathway enrichment analysis. Finally, through an extensive and exact literature search, the biological function of each annotated gene on the island of ROH was determined.

### 2.7. Identification of Consensus Selection Signature Regions in Cashmere Goat Populations

The parameter –homozyg-group was utilized to identify overlapping ROHs (pools) in cashmere goat populations [40,41,42,43]. The output file detailed each ROH of each animal, including their consensus sequences (CONs); their respective union (UNION); and the genome position, size, and number of SNPs. Consensus sequences (CONs) represented common ROH segments shared by at least two animals. In our study, ROH consensus regions with selection signatures in cashmere goats were considered to be homozygous regions (CONs) that overlapped greater than 0.1 Mb and were present in greater than 30% of individuals analyzed in IMC and HSC breeds.

Finally, functional annotation and gene clustering were performed using David and KOBAS (http://www.genome.jp/kegg, accessed on 23 March 2024). GO and KEGG analysis were performed for genes of consensus sequences. The count of significant genes corresponding to each term was identified using a threshold of *p* value ≤ 0.05.

## 3. Results

### 3.1. Population Structure Analyses and Linkage Disequilibrium

In our study, whole-genome resequencing was performed on 55 samples of cash-mere goats from Inner Mongolia and downloaded genome resequencing data for 68 individual goats from the database of NCBI. A total of 123 individual goats from seven populations were included. All 1.66 Tb of raw data were generated, with an averaging depth of 10× for each individual. The principal component analysis (PCA) displayed that the wild goats were separated from the domestic goats (Figure 1A and Appendix A). The zoomed insets in Figure 1A show the PCA for six domestic goat breeds and highlight that these populations were separated based on their utility traits. The plot shows four distinct clusters that are grouped according to their economic traits such as the meat, milk, skin and fiber production of goat populations. The phylogenetic tree (neighbor-joining tree, NJ tree) was similar to the PCA results, and the NJ tree showed all the domestic goat individuals were clearly separated into four categories as wild goats were used as an outgroup (Figure 1B). The two cashmere goat breeds, IMC and HSC, clustered together on the same branch.

The ADMIXTURE analysis indicated K = 5 (cross-validation error = 0.4728) represents the optimal number of discrete genetic populations within the 123 samples (Figure 1C and Appendix A). JNG had the most mixed ancestry of all populations, while HSC had the least. At K  =  5, a majority of IMC individuals separated from HSC individuals.

To further understand the linkage disequilibrium (LD) in domestic and wild goat populations, the LD coefficient (r²) was calculated for all individuals within each population. (Figure 1D). Among the seven populations, the LD plot shows that the IMC population displayed the lowest level of LD, followed by SAA. IBE, while wild goats had the highest level of LD. On the whole, the average r² of IMC declined more rapidly than that observed in the other breeds.

### 3.2. Distribution of Runs of Homozygosity

In total, we identified 57,224 autosomal regions with ROHs. The longest segment (8.5 Mb), containing 305,648 SNPs, was located on chromosome 9, while the shortest segment (0.2 Mb) with 7394 SNPs was identified on chromosome 6 (Appendix A). The largest number of ROHs was detected on chromosome 1; the smallest count was found on 27, and there was an overall decreasing trend in the number of ROHs as the chromosome number increased. In addition, small ROHs were the most common on all chromosomes, followed by medium-sized ROHs, while large ROHs were relatively rare (Figure 2A). IMC had fewer ROHs and a shorter cumulative length of ROHs. Individuals in the IBE population tended to present a higher number of ROHs and greater total lengths of ROHs compared to other populations (Figure 2B). The distribution of the total ROH length across different populations is shown in Figure 2C. JNG and IMC showed a distribution with lower overall lengths of ROHs compared to the IBE population. But HSC displayed a longer total length than IMC. The ALG population had a relatively high median total length of ROHs. This suggested that this population contains a larger number of individuals with longer segments of ROHs. The BOE population with a wide distribution exhibited significant variation in the ROH lengths among individuals within this population. It was evident that the small and medium ROH length class had a relatively high mean number of ROHs across all populations (Figure 2D). The large ROH length class showed a markedly reduced mean number of ROHs. IMC showed the lowest mean numbers of ROHs across all length classes. In contrast, the IBE population had the largest mean number of ROHs in all classes, followed by HSC, indicating a possible high level of genomic homozygosity. Table 2 and Appendix A summarize the fundamental statistics of the three ROH categories. The analysis of the different ROH length classes for each population showed that ROH segments ranging from 0.3 to 1.5 Mb were the most common, while the long ROH segments (>1.5 Mb) accounted for just 1.55% of all three classes. This indicated that medium–long segments covered the highest proportion of the genome (66.59%).

### 3.3. Inbreeding Coefficients

To evaluate the inbreeding extent for each goat population, the population inbreeding coefficients and the individual inbreeding coefficients were calculated (Table 3 and Appendix A). $F_{roh}$ in the seven populations ranged from 0.0263 to 0.4780. Compared to other domestic goat populations, the IMC population exhibited the lowest inbreeding coefficient (0.0263), which is similar to the previous study [29]. Contrastingly, the IBE population, a wild goat population, had the highest inbreeding coefficient (0.4780) and showed an exceptionally high total ROH length (11,788.44 Mb), indicating a higher level of inbreeding within this population. Thid was consistent with the findings of previous studies, where the modern wild bezoar from Iran displayed the highest extreme $F_{roh}$ [44,45]. HSC had a relatively higher inbreeding coefficient compared to other domestic breeds. Other populations, such as JNG, SAA, ALG and BOE, exhibited varying degrees of inbreeding coefficients, ranging from 0.0446 to 0.0708.

### 3.4. ROH Islands of Domestic Goat Populations

ROH islands refer to regions of runs of homozygosity that are shared among multiple individuals within a population. These regions may be preserved or enhanced in the population due to factors such as genetic drift, natural selection or artificial selection.

Seventy-four ROH islands were identified with 10,839,351 SNPs across the 29 autosomes based on the top 0.1% of SNPs for ROH incidence. Among them, 50 genes were detected as putative candidate genes (Table 4, Appendix A). The Manhattan plot showed the incidence of SNPs in ROHs per chromosome in HSC and BOE (Figure 3) and other populations (Appendix A). The HSC goat population displayed higher ROH incidence levels (>30%), with numerous ROH islands exceeding 80% incidence. In contrast, the BOE goat population showed generally low ROH incidence, with seven notable ROH islands on chromosomes 8, 11, 13, 16 and 19, respectively.

KEGG and GO enrichment analysis was conducted for 50 genes identified as potential candidate genes (Appendix A). The GO analysis revealed seven GO entries were significantly enriched, such as the positive regulation of cell proliferation (GO:0008284), the negative regulation of granulocyte differentiation (GO:0030853) and nuclei (GO:0005634). Additionally, the enrichment analysis of KEGG just identified two pathways, including the Ras signaling pathway and cell-adhesion molecules. Notably, based on previous studies, several genes may be associated with the economic traits of goats. For instance, they were related to the meat trait (*FGF1*, *CFAP74*) fiber (*RERE*, *PTPRM*, *FGF5*) and milk (*GRID2*, *RARA*) production, fertility (*BIRC6*, *ECE2*, *CDH23*, *PAK1*), disease or cold resistance and adaptability (*PDCD1LG2*, *SVIL*, *PRDM16*, *RFX4*, *SH3BP2*, *PTPRM*) and body size and growth (*TMEM63C*, *SYN3*, *SDC1*, *STRBP*, *SMG6*, *ACYP2*, *RFX4*) (Table 4).

### 3.5. The Consensus ROH of IMC and HSC Breeds

To identify genes related to cashmere traits, ROH consensus sequences for the IMC and HSC breeds were explored. A total of 135 consensus ROH pools were identified (Appendix A). The gene annotation of consensus ROHs revealed 191 protein-coding genes (Appendix A). Notably, a specific consensus ROH (115.36 kb: 95,438,689–95,554,048 bp) contained the *FGF5* gene on chromosome 6, which was shared by 19 individuals from the IMC breed. For the HSC population, a consensus ROH (106.374 Kb: 14,880,801–14,987,174 bp) was identified on chromosome 11, which was common for 13 individuals.

Enrichment analysis revealed that genes within consensus ROH regions significantly contributed to 39 GO terms and 60 KEGG pathways (Figure 4, Appendix A). Notably, the GO terms included cell division (GO:0051301), nucleoplasm (GO:0005654), nucleus (GO:0005634), centrosome (GO:0005813), actin cytoskeleton (GO:0015629) and protein binding (GO:0005515), which were involved in cell division and cell metabolism. We deliberated whether these GO terms may be related to the morphogenesis of hair follicles. KEGG pathway analysis revealed that 60 pathways were significantly enriched, such as MAPK signaling, Ras signaling, PI3K-Akt signaling, Rap1 signaling, JAK-STAT signaling, Notch signaling, metabolic pathways, melanogenesis pathways, focal adhesion and signaling pathways regulating the pluripotency of stem cells, crucial for the regulation of hair follicle development. In the above GO terms and pathways, it had been confirmed that the included genes, *FGF5*, *KIT* [46], *DVL3* and *NRAS*, were crucial for cashmere fibers of cashmere goats.

## 4. Discussion

### 4.1. Population Structure and Linkage Disequilibrium

In our study, we conducted a population structure analysis and LD on a total of 123 individual goats from seven populations. The population structure analysis indicated a consistent clustering pattern based on PCA, NJ phylogenetic trees and admixture analyses. It has been noted that in the experimental samples, apart from the wild goat population, IBE, the other six domestic goat populations were clustered into four groups, which is generally in agreement with their economic traits [47].

LD, a significant genetic concept, refers to the association of alleles at distinct loci that are not random. Insights into the history of a population and evolution can be observed from diminishments in LD between genetic markers. The LD decay pattern among genetic markers offers insightful observations on the history of a population and evolution. A faster rate of LD decay indicates higher genetic diversity, correlating with phenomena like genetic drift, migration and selection within a population. Overall, IMC exhibited higher genetic diversity, indicating the conservation measures employed on farms for IMC were successful. Likewise, the IBE population exhibited extensive LD, suggesting the possibility of inbreeding within this population. It is significant to enhance the preservation of these populations’ genetic resources to avoid diminishing genetic diversity. Thus, it is necessary to relentlessly explore genetic variation and structure. This is key to averting a swift reduction in diversity, which is vital for sustainable improvements in the livestock and poultry industry.

### 4.2. Patterns of Runs of Homozygosity (ROHs)

The analysis of ROHs throughout seven goat populations unveiled ROH patterns (Figure 2A,B). Some goat breeds, like IMC and JNG, characterized by shorter and fewer ROHs, indicate a relatively higher genetic diversity. In contrast, long ROH segments were observed in certain populations, particularly in the IBE and HSC populations, suggesting historical inbreeding or that a reduced effective population size might have occurred. Longer ROH segments suggested recent inbreeding, whereas shorter ROH segments implied more distant inbreeding events [1,48]. In our research, the IMC population mainly showed short ROH segments, whereas the proportion of long segments was very small, particularly those exceeding 1.5 Mb. This pattern indicated that the IMC population had a low level of inbreeding, with ancient ancestors being the primary group affected by inbreeding events. Therefore, compared to wild goats and other domestic goat breeds, the IMC population may display reduced inbreeding occurrences and enhanced genetic diversity, which were similar with the result of genetic diversity and the LD analysis. Predominantly, ROH segments shorter than 1.5 Mb were the most common, aligning with findings from preceding research on livestock and poultry [41,42,45,49,50,51].

### 4.3. Inbreeding Levels within Populations

The assessment of runs of homozygosity (ROHs) and inbreeding coefficients across the studied goat populations has revealed notable differences in inbreeding levels, which are crucial for understanding the genetic diversity of these populations.

Inbreeding coefficients estimated by ROHs ranged from 0.0263 to 0.4780. The inbreeding coefficients across different populations could reflect the breeding practices and historical management of these breeds. The IMC population exhibited the lowest inbreeding coefficient of 0.0263, which was generally consistent with previous research [29]. This could be attributed to diverse breeding practices or a larger effective population size, which is essential for the long-term sustainability of the breed. In contrast, the IBE population showed an inbreeding coefficient of 0.4780. This was similar to previous studies that reported the highest rates of extreme $F_{roh}$ in modern wild bezoar from Iran [44,45] and diminished diversity levels relative to domestic goats [52]. HSC had a relatively higher inbreeding coefficient compared to other domestic goat populations. This high level of inbreeding is a cause for concern, as it indicates a limited genetic pool and potential risks of inbreeding. The government should develop and implement a variety of management plans, as well as selective breeding programs, to detect and manage the genetic diversity of populations. Other populations, including SAA, JNG and ALG, displayed varying degrees of inbreeding coefficients, ranging from 0.0446 to 0.0708. These values suggest moderate levels of inbreeding, which could be reflective of controlled breeding strategies aimed at maintaining specific traits.

### 4.4. Functional Enrichment Analysis

In the ROH islands of domestic goat populations, several genes were identified, such as *FGF1*, *RERE*, *PTPRM*, *FGF5*, *GRID2*, *RARA* and *BIRC6* (Table 4). Seven GO terms and two KEGG pathways of these genes were significantly enriched. For instance, GO terms included the positive regulation of cell proliferation (GO:0008284), the negative regulation of granulocyte differentiation (GO:0030853) and nuclei (GO:0005634). KEGG pathways were enriched, including the Ras signaling pathway and cell-adhesion molecules. Through an extensive and precise literature search, we determined the function of each annotated gene on the ROH island. In previous research, these genes were mainly related to meat, milk and fiber production; reproductive capacity; and disease resistance. The genes that we found in HSC and IMC, cashmere goat breeds, were related to fiber traits, for instance, *RERE*, *PTPRM* and *FGF5*. *RERE* was related to the wool staple length in Akkaraman lambs [53]. *PTPRM* is known as a hair-related gene that regulates cell–cell communication in keratinocytes [54,55]. But it is also associated with brucellosis resistance in sheep and muscling in pig [56,57,58]. In addition, we also found some genes were related to adaptability (*RFX4*) [59] and immunity (*SH3BP2*) [60] in cashmere goats. These genetic factors may potentially improve the adaptability of cashmere goats in cold environments as well as their resistance to pathogens. ALG is a dairy goat, and we have identified *GRID2* as a gene related to milk production. *GRID2* (glutamate ionotropic receptor delta-type subunit 2) is linked to the central suspensory ligament in Chinese Holstein cows [61]. It is strongly related to dry matter intake, average daily gain, birth weight and milk fat yield in cattle [62]. However, earlier research indicated that *GRID2* plays a role in the sexual maturity of Simmental cattle, the temperament trait of sheep and the litter size of goat [63,64,65]. On BOE, key genes associated with traits such as fertility (*BIRC6*), disease resistance (*PDCD1LG2*, *SVIL*), cold tolerance (*PRDM16*), bone development (*ACYP2*), meat production (*CFAP74*) and growth performance (*SMG6*) were discovered. It was also linked to the introduction and improvement of Boer goats. *BIRC6* (baculoviral inhibitors of apoptosis repeat-containing 6) has been confirmed to be involved in the development of follicles in broilers and early embryonic development and fertility in *Bos indicus* [66,67,68]. *PDCD1LG2* (*PDCD1* ligand 2) and *SVIL* are immunostimulatory genes. It has been reported that *PDCD1LG2* in Bohuai goats and *SVIL* in *Bos indicus* are related to immunity [69,70]. *PRDM16* is key in regulating the formation of brown fat cells and the production of brown fat [71,72,73]. PRDM16 is also related to cold tolerance. For example, in sheep, hypothermia due to cold exposure at birth was prevented by the speedy activation of non-shivering thermogenesis in brown adipose tissue [74]. In addition, it has been discovered that *CFAP74* is associated with meat production in goats and yearling fiber diameter in sheep [53,75]. In SAA, apart from the genes related to milk production traits (*RARA*), the majority of the annotated genes are associated with embryonic development or litter size, such as *XRCC4* [76], *CDH2*3 [77] and *PAK1* [78,79]. Consistent with previous studies, *RARA* was found on a region of chromosome 19 that displayed a negative pleiotropic impact on milk production traits (milk, fat yield and protein yield) as well as udder development (udder floor position and rear udder attachment) and was also linked to responses to intramammary infections [80].

The functional enrichment analysis of genes identified in consensus ROH regions provided valuable insights into the biological processes and pathways potentially influencing economically important traits in cashmere goats. In our research, the analysis of KEGG pathways indicated that several significant pathways were enriched in relation to fiber traits. These pathways included focal adhesion, JAK-STAT signaling, Ras signaling, PI3K-Akt signaling, Rap1 signaling, Notch signaling, metabolic pathways, melanogenesis pathways, MAPK signaling and signaling pathways regulating the pluripotency of stem cells. These pathways are crucial in regulating the development of hair follicles. The MAPK signaling pathway was recognized for promoting the growth and differentiation of hair follicle cells, fostering their regular developmental cycle and promoting skin-derived precursors by enhancing both the proliferation and hair-inducing capacity [81]. Notch signaling, Ras signaling and Rap1 signaling are important to the in vitro proliferation of hair follicle stem cells [82]. PI3K-Akt signaling plays a necessary role in the new formation of hair follicles [83]. The JAK-STAT signaling pathway is known to be related to the regulation of hair follicle growth, and it has been targeted for the purposes of alopecia areata in the clinical setting [84]. Focal adhesions have been detected during the vital transition from the telogen to the anagen phase, potentially acting as key biomarkers for the regeneration process between these hair growth stages [85,86]. Signaling pathways regulating the pluripotency of stem cells play an important part in determining cell fate by controlling self-renewal and diversification into various cell types [87]. Melanogenesis is an important biochemical process in which melanocytes produce melanin, a crucial element involved in the formation of coat color in mammals [88,89]. In our study, *DVL3*, *NRAS* and *KIT* in the melanogenesis pathway were identified as potential candidate genes influencing cashmere goat fiber color. Previous studies have reported on the involvement of *NRAS* and *KIT* in this process [90]. Mutations or deletions of *KIT* can result in specific hair and skin colors among mammals [91,92,93]. In the Rap 1 and MAPK pathways, we identified the *FGF5* gene. *FGF5* (fibroblast growth factor 5) is notable as it inhibits hair elongation and is related to hair growth and length in mammals [94,95,96]. These findings not only enhance our understanding of the genetic basis of these characteristics but also pave the way for targeted genetic improvement.

## 5. Conclusions

In conclusion, our study explored ROHs through seven goat populations’ genomes and counted the inbreeding coefficient to assess inbreeding levels. Additionally, we pinpointed ROH islands that harbor genes associated with economically vital traits. Our findings indicated that historical inbreeding has affected the IMC population, which exhibits a relatively low inbreeding level. Through analyzing regions identified in ROH islands and consensus, we discovered certain genes associated with economically crucial traits, including meat, fiber and milk production; fertility; growth; as well as the resistance and adaptability of goats. Our study contributes to a more comprehensive understanding of genetic diversity, the level of inbreeding and essential genes potentially under selection. It also offers a valuable perspective on future conservation measures and the use of cashmere goats.

## Figures and Tables

**Figure 1 animals-14-01246-f001:**
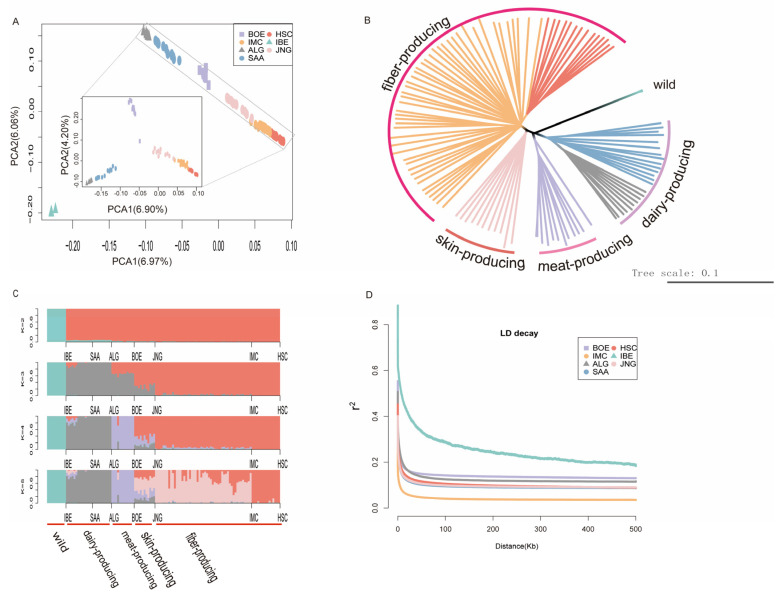
Population genetics structures analyses and linkage disequilibrium. (**A**) Principle component analysis (PCA); each point represents a single individual. The colors in B and D represent the same groups as indication in the legend of A. (**B**) Phylogenetic tree constructed by the neighbor-joining (NJ) method. (**C**) Population structure plot of seven goat populations at K = 2–5, (**D**) LD decay map measured by r² over distance between SNPs in seven populations.

**Figure 2 animals-14-01246-f002:**
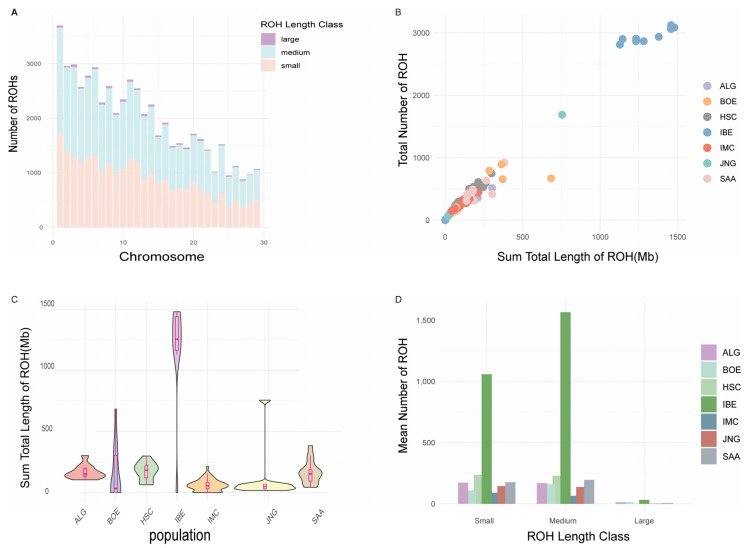
The distribution of ROHs identified in different populations across autosomes. (**A**) This graph is a bar chart showing the number of ROHs on different chromosomes. ROHs are divided into three length classes: small (0–0.3 Mb), medium (0.3–1.5 Mb) and large (>1.5 Mb). (**B**) Relationship between the number of runs of homozygosity (ROH) per individual and the total length of the genome covered by them. The x-axis shows the sum total length of ROHs (Mb), and the y-axis shows the total number of ROHs. Every circle represents a different individual within a population, with the groups labeled as ALG, HSC, IMC and so on. (**C**) Violin plots and boxplots of sum total length of ROHs (SROHs). The form of each violin reflects the distribution density of cohort, with wider sections of the violin plot indicating higher frequency of observation at that y -value. Inside each violin, there is a boxplot that shows the median, the interquartile range (the length of the box) and the outliers (the points outside the whiskers). (**D**) The descriptive statistics for the ROHs, categorized by ROH length class across different populations, are presented as mean counts of ROHs (Y-axis) by class of ROH length.

**Figure 3 animals-14-01246-f003:**
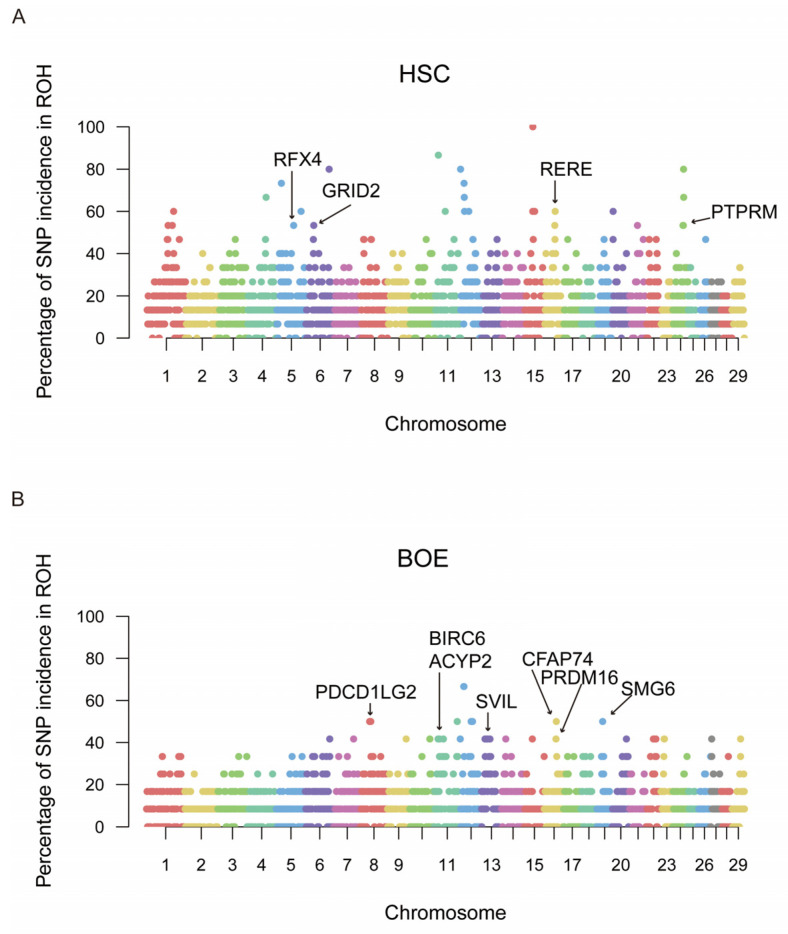
Manhattan plot of SNPs in ROHs for HSC (**A**) and BOE (**B**). The x-axis exhibits positions along each chromosome.

**Figure 4 animals-14-01246-f004:**
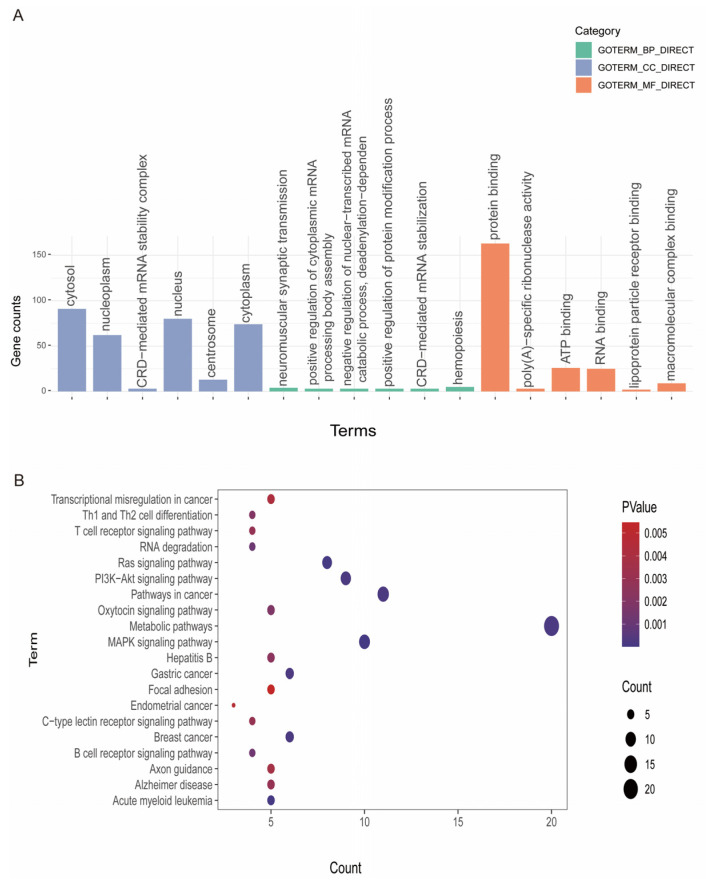
(**A**) A Gene Ontology (GO) functional enrichment analysis. (**B**) KEGG enrichment pathways of gene annotation in the consensus ROHs of IMC and HSC breeds.

**Table 1 animals-14-01246-t001:** Information about the goat populations in this study.

Code	Breed/Population	Sample Size	Location	Use
IMC	Erlangshan cashmere goat	40	Bayannur City, Inner Mongolia, China	Cashmere and meat
IMC	Albas cashmere goat	4	Ordos City, Inner Mongolia, China	Cashmere and meat
IMC	Alxa cashmere goat	7	Alxa League, Inner Mongolia, China	Cashmere and meat
HSC	Hanshan White cashmere goat	15	Chifeng City, Inner Mongolia, China	Cashmere and meat
JNG	Jining Gray goat	11	Jining City and Heze City, Shandong Province, China	Lambskin and meat
ALG	Alpine goat	10	France	Milk
SAA	Saanen Dairy goat	8	South Korea	Milk
SAA	Saanen Dairy goat	6	The United Republic of Tanzania	Milk
BOE	Boer goat	3	South Korea	Meat
BOE	Boer goat	3	New Zealand	Meat
BOE	Boer goat	6	Australia	Meat
IBE	Siberian Ibex	10	Switzerland	

**Table 2 animals-14-01246-t002:** Summary statistics for the numbers and lengths (in Mb) of ROHs based on different ROH length classes (0–0.3 Mb, 0.3–1.5 Mb and >1.5 Mb).

ROH Length (Mb)	ROH Number	Number Percentage (%)	Total_Length (Mb)	Mean ± SD (Mb)	Length Percentage (%)
0–0.3	25,933	45.32	6280.59	0.24 ± 0.028	25.64
0.3–1.5	30,406	53.14	16,313.46	0.54 ± 0.24	66.59
>1.5	885	1.55	1905.19	2.15 ± 0.81	7.78

**Table 3 animals-14-01246-t003:** Population name, sample size (*n*, the number of samples with a count of ROH greater than zero), the length of ROHs and inbreeding coefficient based on ROHs ($F_{roh}$).

Population	*n*	Total ROH Length (Mb)	Inbreeding Coefficient
IMC	47	3045.20	0.0263
HSC	15	2619.12	0.0708
JNG	11	1209.02	0.0446
SAA	14	2326.06	0.0674
ALG	10	1672.71	0.0678
BOE	12	1838.68	0.0621
IBE	10	11,788.44	0.4780

**Table 4 animals-14-01246-t004:** Candidate genes located in genomic regions with a high frequency of ROHs.

Population	Chromosome	Position (Mb)	Gene Name	Gene Function
ALG	6	32.73~59.46	*GRID2*	Udder development, growth, fertility, temperament traits
	7	56.81~57.00	*FGF1*	Adipocyte differentiation
	10	14.63~30.13	*TMEM63C*	Body size and development
			*MNAT1*	Cell cycle and DNA repair
BOE	8	38.47~39.00	*PDCD1LG2*	Disease resistance
	11	15.04~21.07	*BIRC6*	Follicular development, fertility
			*ARHGEF33*	Retinal development
	11	36.48~37.00	*ACYP2*	Growth
			*NEBL*	Puberty
	13	34.05~35.34	*SVIL*	Disease resistance
	16	47.76~48.17	*PRDM16*	Formation of brown fat cells, cold resistance
	16	49.30~50.00	*CFAP74*	Meat production, staple length
	19	22.96~23.00	*SMG6*	Growth
	22	37.21~38.13	*PRICKLE2*	Postpartum dysgalactia syndrome
HSC	5	18.57~68.73	*RFX4*	Adaptability, body size and development, fertility, udder development and milk production
	6	32.54~95.45	*GRID2*	Udder development, growth, fertility, temperament traits
	11	14.88~15.00	*BIRC6*	Follicular development, fertility
	15	31.98~32.19	*STIM1*	Pulmonary circulation, body size and development, meat production, neural function or behavior
	16	41.63~43.22	*RERE*	Staple length
	24	41.14~42.58	*PTPRM*	Interaction between keratinocytes, meat production, immune
			*PIEZO2*	Meat production
IMC	1	82.57~108.78	*ECE2*	Fertility, embryo development
	6	95.44~116.63	*FGF5*	Hair growth and length
			*SH3BP2*	Immune
	11	14.98~15.04	*BIRC6*	Follicular development, fertility
	15	31.97~32.19	*STIM1*	Pulmonary circulation, body size and development, meat production, neural function or behavior
	19		*SMG6*	Growth
JNG	1	109.15~132.19	*STAG1*	Embryonic development
SAA	5	59.27~69.98	*SYN3*	Body size and development, meat production
	5	70.00~98.96	*SYN3*	Body size and development, meat production
	7	27.59~57.47	*XRCC4*	Embryonic development
	7	58.31~59.89	*MATR3*	Corpus luteum
	8	74.99~75.00	*NOL6*	Skin color
	11	78.35~93.90	*SDC1*	Body size and development, meat production, cashmere fineness
			*STRBP*	Body size
	19	24.51~40.23	*RARA*	Milk production, udder development, hair follicle morphogenesis
	28		*CDH23*	Fertility
	29	17.94~18.00	*PAK1*	Fertility, puberty

## Data Availability

The data presented in this study are available upon request from the corresponding author.

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
