# Peer review of "Genomic Inbreeding and Runs of Homozygosity Analysis of Cashmere Goat"

_animals, 2024, doi:10.3390/ani14081246_

Round 1
Reviewer 1 Report
Comments and Suggestions for Authors
Manuscript Number: animals-2958905
Title: Genomic inbreeding and runs of homozygosity analysis of cashmere goat
General comments and judgment
The Authors aimed to characterize genome wide ROH patterns and inbreeding levels; and to identify ROH islands and putative candidate genes associated with cashmere trait in the cashmere goat based. The study offers the opportunity to reflect on the future breeding, in order to assess inbreeding levels, maximizing the conservation and utilization of goat breeds. The manuscript seems to be well written, in fluent English language: clear, precise, and easy to understand. Aims are well addressed, and results adequately discussed.
However, some points remain to be clarified, especially in the Materials and Method section. Furthermore, some small correction may be suggested, as specified below.
Main comments
- What were the criteria for the selection of the studied population? Is there a specific reason or is it just a coincidence that you had samples of those goat breeds?
- Please add some additional information in the M and M section on the enrolled cashmere breeds. This study refers to different cashmere populations (IMC – three types? – and HSC), without adequate presentation. It would be useful for the reader to have some more data on these different populations and what this difference consists of (maybe geographical, morphological, ethnic? We don’t know).
- Figures: the quality of all the figures does not seem to be very good (when enlarged they become grainy and do not allow adequate reading). Please check and improve if necessary.
Minor comments
Line 33: please explain all the acronyms at their first appearance in the text.
Lines 50-51: the sentence sounds incomplete, please check.
Lines 76-77: please rephrase.
Line 82: the term “Erlangshan” compares only here and in Table 1. What is it referring to? Please explain or remove it.
Line 96: please reverse the order: name of the breed first and acronym in brackets, as for the others.
Line 221: please note that Supplementary Table S1 is in a different format than the others and does not open, please check, and correct.
Author Response
We gratefully thank the reviewer for making their constructive remarks and useful suggestions, which can significantly improve the manuscript. Each suggested revision and comment, brought forward by the reviewers was accurately incorporated and considered.
Comments 1: What were the criteria for the selection of the studied population? Is there a specific reason or is it just a coincidence that you had samples of those goat breeds?
Response 1: Thank you for your attention to our study and for the opportunity to clarify the selection criteria for our studied populations.
Our research focuses on genomic inbreeding and runs of homozygosity analysis in cashmere goats, with a specific interest in traits such as cashmere traits, adaptability and cold resistance. To this end, we have carefully selected four cashmere goat populations known for their superior traits in these aspects: Inner Mongolia Cashmere Goat and Hanshan White Cashmere Goat.
To provide a comprehensive perspective on the genomic architecture and to enable a robust comparison, we also included non-cashmere goat populations in our study. These include: Jining Grey Goat (use: skin, China), Alpine Goat (use: milk, France), Saanen Dairy Goat (use: milk, South Korea and United Republic of Tanzania) and Boer Goat(use: meat, South Korea, New Zealand and Australia). Additionally, a wild goat population was incorporated as an outgroup for comparative purposes.
Comments 2: Please add some additional information in the M and M section on the enrolled cashmere breeds. This study refers to different cashmere populations (IMC – three types? – and HSC), without adequate presentation. It would be useful for the reader to have some more data on these different populations and what this difference consists of (maybe geographical, morphological, ethnic? We don’t know).
Response 2: We have revised this point and added detailed content to the M and M section.
Comments 3: Figures: the quality of all the figures does not seem to be very good (when enlarged they become grainy and do not allow adequate reading). Please check and improve if necessary
Response 3: Sure. We have already converted all figures to TIF format, which makes figures more clear
Comments 4: Line 33: please explain all the acronyms at their first appearance in the text.
Response 4: We have revised “IMC” in line 33 to “Inner Mongolia Cashmere goat”.
Comments 5: Lines 50-51: the sentence sounds incomplete, please check.
Response 5: We have revised this sentence.
Comments 6: Lines 76-77: please rephrase.
Response 6: We have revised this sentence.
Comments 7: Line 82: the term “Erlangshan” compares only here and in Table 1. What is it referring to? Please explain or remove it.
Response 7: we have removed “Erlangshan” in line 82.
Comments 8: Line 96: please reverse the order: name of the breed first and acronym in brackets, as for the others.
Response 8: We have revised the format.
Comments 9: Line 221: please note that Supplementary Table S1 is in a different format than the others and does not open, please check, and correct.
Response 9: We have revised the format of Supplementary Table S1 and the new table was included in the attachment files.
Reviewer 2 Report
Comments and Suggestions for Authors
Thank you for your painstaking work! I didn't see any significant flaws. There are a couple of clarifications for improvement: It is not entirely clear whether LD (indep-pairwise) filtering has been done? (line 138) It would be great to have a table like this (Tab 2) in the supplementary materials for each breed. (line 259) Figure 4A is not readable, could the authors enlarge the font or flip the image?
Author Response
We gratefully thank the reviewer for making their constructive remarks and useful suggestions, which can significantly improve the manuscript. Each suggested revision and comment, brought forward by the reviewers was accurately incorporated and considered.
Comments 1: It is not entirely clear whether LD (indep-pairwise) filtering has been done? (line 138)
Response 1: Thank you for your attention to our study. We have not performed LD filtering for the LD decay analysis
Comments 2: It would be great to have a table like this (Tab 2) in the supplementary materials for each breed. (line 259
Response 2: Thanks for your suggestion. We have added Supplementary Table S2 to describe the basic statistical data of different length categories for each population. And we have updated the numbering of the Supplementary Table and uploaded the new supplementary table files
Comments 3: (line 259) Figure 4A is not readable, could the authors enlarge the font or flip the image?
Response 3: Sure. We have already revised this figure.

Reviewer 3 Report
Comments and Suggestions for Authors
The main aims of the paper are, on one hand, to characterize genome-wide ROH patterns and inbreeding levels and, on the other hand, to identify ROH islands and putative candidate genes associated with cashmere trait in the cashmere goat based on resequencing data from Inner Mongolia Erlangshan Cashmere goats, Hanshan White Cashmere goats and other non-cashmere goats.
The scientific content of the present article is very high, being very well documented and the most modern and up-to-date technologies were applied.
The applied methodology is at a high scientific level in the field.
The materials and methods chapter includes very modern and state-of-the art molecular genetics methods and statistics.
The reference list is exhaustive, and includes the most important publication in the field.
The manuscript is well-structured and the discussions are comparatively emphasizing the obtained results and the scientific knowledge in the field.
The results are well presented, together with the graphics which better represents the obtained results, also complemented by the supplementary files.
The final conclusions are based on the obtained results, are well presented and synthetized. The present study explored ROH through the seven goat population’s genome and counted the inbreeding coefficient, in order to assess the inbreeding levels.
Main findings indicated that historical inbreeding has affected the IMC population, which exhibits a relatively low inbreeding level. By analyzing regions identified in ROH island and consensus, there were discovered different genes associated with economically crucial traits, such as: meat, fibres and milk production, fertility, growth, as well as resistance and adaptability of goat.
The animal study protocol was made according to Ethical approval for animal survival provided by the animal ethics committee of the Institute of Animal Science, Chinese Academy of Agricultural Sciences (IAS-CAAS) with the following approval number: IAS2019-61.
I would suggest to the authors to revise again the English editing of the text.
Comments on the Quality of English Language
I would suggest to the authors to revise again the English editing of the text.
Author Response
We gratefully thank the reviewer for making their constructive remarks and useful suggestions, which can significantly improve the manuscript. Each suggested revision and comment, brought forward by the reviewer was accurately incorporated and considered.
Point 1: I would suggest to the authors to revise again the English editing of the text
Response 1: We have revised the English editing once more in accordance with the suggestions from the reviewers